# Hemostatic Abnormalities in Gaucher Disease: Mechanisms and Clinical Implications

**DOI:** 10.3390/jcm11236920

**Published:** 2022-11-24

**Authors:** Silvia Linari, Giancarlo Castaman

**Affiliations:** Center for Bleeding Disorders and Coagulation, Department of Oncology, Careggi University Hospital, 50134 Florence, Italy

**Keywords:** Gaucher disease, bleeding, hemostasis, platelets, clotting factors

## Abstract

Gaucher disease (GD) is a rare inherited lysosomal metabolism disorder, characterized by an accumulation into lysosomes of reticuloendothelial cells, especially in the bone marrow, spleen, and liver of β-glucosylceramide and glucosyl sphingosine, which is its deacylated product. Impaired storage is responsible for a chronic inflammatory state at the sites of accumulation and together represents the pathophysiological cause of GD. GD is a progressive, multi-organ chronic disorder. Type 1 GD is the most prevalent form, with heterogeneous multisystem involvement and different severity of symptoms at any age. Hematological involvement is consistent, and a bleeding tendency is frequent, particularly at diagnosis. Several coagulation and primary hemostasis abnormalities are observed in GD. Bleeding manifestations are rarely severe and usually mucocutaneous. Post-operative, delivery, and post-partum hemorrhages are also common. Thrombocytopenia, platelet function defects, and clotting abnormalities, alone or variably associated, contribute to increase the risk of bleeding in GD. Enzyme replacement therapy (ERT) or substrate reduction therapy (SRT) are the two specific available treatments effective in improving typical hematological symptoms and abnormalities, including those of hemostasis. However, the use of medication to potentiate hemostasis may be also useful in defined clinical situations: recent starting of ERT/SRT, surgery, delivery, and life-threatening bleeding.

## 1. Introduction

Gaucher disease (GD) is an autosomal recessive lysosomal storage disease that is caused by deficiency of the enzyme β-glucocerebrosidase (β-GCase), which is required for the degradation of glycosphingolipids. Deficiency of β-GCase is responsible for the accumulation of glucosylceramide and its deacylated product glucosylsphingosine into lysosomes of reticuloendothelial cells [1,2]. These lipid-laden cells are known as Gaucher cells. Gaucher cells are large and characterized by eccentric nuclei, condensed chromatin and cytoplasm with heterogeneous “crumpled tissue paper” [2]. The bone marrow, spleen and liver are particularly infiltrated by these cells in GD, leading to the main clinical signs of the disease at diagnosis [3].

The GD phenotypic pattern is highly variable [4], from asymptomatic forms to perinatal-lethal forms, but three major clinical phenotypes have been identified by the absence (type 1) or presence (types 2 and 3) of primary central nervous system (CNS) involvement (Table 1). However, it is now recognized that neuropathic GD represents a phenotypic continuum, ranging from extrapyramidal syndrome in type 1 to hydrops fetalis in type 2 [5]. The most prevalent form of the disease is type 1, which covers almost 95% of Caucasian patients [6]. Type 1 GD is characterized by enlargement of the liver and/or spleen, thrombocytopenia, anemia, and skeletal abnormalities. Clinical or radiological evidence of bone disease occurs in 70–100% of type 1 patients. Bone disease ranges from asymptomatic osteopenia to focal lytic or sclerotic lesions and osteonecrosis [7]. Bone involvement may also lead to acute or chronic bone pain, pathologic fractures, and subchondral joint collapse with secondary degenerative arthritis, often being the most debilitating manifestation of type 1 GD. The lung is a rarer organ involvement of type 1 GD, with interstitial disease and pulmonary hypertension. Neurologic complications (spinal cord or nerve root compression) may occur as the consequence of bone disease (e.g., severe osteoporosis with vertebral compression) or coagulopathy [8], although type 1 GD does not directly cause primary CNS disease. A sensory motor axonal polyneuropathy was diagnosed in 10.7% of 103 patients followed prospectively for two years [9]. Finally, in type 1 GD, a significant risk of severe hematological malignant disorders, mainly multiple myeloma and B-cell lymphoma, has been also reported [10,11,12,13]. Gammopathies, multiple myeloma, and B-cell lymphoma are felt to be the result of an atypical immune activation [14].

GD is the first lysosomal storage disorder for which an effective therapy has become available. Two approaches are possible for patients with type 1 GD: enzyme replacement therapy (ERT) and substrate reduction therapy (SRT). ERT, available since 1991, is based on the provision of sufficient exogenous enzyme to overcome the block in the catabolic pathway. It has been shown to be effective in reducing glucosylceramide and glucosylsphingosine storage burden and the deleterious effects caused by their accumulation. ERT is effective in counteracting the peripheral symptoms, in particular hepatosplenomegaly and hematological symptoms [15,16,17,18,19]. Three different human recombinant GCases have been approved. Two are available in EU and USA: Imiglucerase (Cerezyme, Genzyme Corporation, Cambridge, MA, USA) and Velaglucerase alfa (VPRIV, Shire HGT, Cambridge, MA, USA). An additional agent, Taliglucerase alfa (Elelyso, Protalix Biotherapeutics, Carmiel, Israel), is not available in the EU. ERT is well tolerated. Approximately 10–15% of patients develop antibodies to infused imiglucerase, whereas antibodies to velaglucerase have been reported in only 1% of cases. However, these anti-drug antibodies are not neutralizing and do not influence the efficacy of replacement treatment.

SRT is an alternative treatment strategy that seeks to balance glucosylceramide production and its impaired rate of degradation by partly inhibiting glucosylceramide synthase [20]. SRT aims to restore metabolic homeostasis by limiting the synthesis of the substrate precursor to a level that can be effectively cleared by the mutated enzyme endowed with residual hydrolytic activity [21]. The first attempt of an SRT for GD with the iminosugar N-butyldeoxynojirimycin, Miglustat (Zavesca, Actelion Corp, Allschwil, Switzerland) [22] did not achieve the expected results due to its low benefit–risk profile. The new ceramide analog of the substrate, Eliglustat (Genz-112638; Genzyme Corp) [23], shows a higher potency and improved safety than Miglustat. Several studies showed its effectiveness in achieving and maintaining the therapeutic goals for visceral and hematological manifestations in naïve GD1 patients and in those who switched from ERT, with a good safety profile [24,25,26,27,28].

## 2. GD and Inflammation

The multi-organ diffuse infiltration by Gaucher cells alone cannot explain the multiple and heterogeneous manifestations of the disease. The accumulation of the β-GCase substrate leads to a secondary activation of macrophages associated with an autophagy disruption and an onset of a cascade of inflammatory events that further worsen the disease [29,30,31,32].

Increased expression of tumor necrosis factor-α (TNF-α interleukin-1β (IL-1β)), interleukin-1 receptor antagonist, soluble interleukin-2 receptor (sIL-2R), interleukin-6 (IL-6), interleukin-8 (IL-8), interleukin-10 (IL-10), CD14, CD163s, MIP-1β, TGFβ, M-CSF, CCL-18, and chitotriosidase have been found in patient plasma [33,34,35,36,37,38].

The pathogenetic mechanisms of inflammatory propagation in GD have yet to be delineated, but a possible pathway has been proposed [39]. The primary Gaucher cells activation triggers the release of monocytes (MOs) and polymorphonuclear neutrophils (PMNs) recruiting cytokines and chemokines. Monocyte chemoattracting protein-1 (MCP-1) recruits circulating MOs, whereas IL8 and TNF-a induce the PMNs migration into the different visceral organs. With migration from the blood into the visceral organs, MOs mature into tissue-specific macrophages with GCase defect, leading to an increase in Gaucher cells and further release of IFN-g, IL-4, IL-6, and TGF-b. IFN-g and IL-4 cause the development of T helper-1 (Th1) and Th2 cell-mediated responses, whereas IL-6 induces the development of T-follicular cells, thus activating B-cells in germinal centers with associated hypergammaglobulinemia. IL-6 together with TGF-b impact Th17 cell development with IL-17 production and subsequently that of IL-8 with recruitment of circulating PMNs in GD visceral organs. In addition to IL-8, Gaucher cells also secrete IL-1b, INF-g, and TNF-a, which are crucial for recruiting PMNs with subsequent release of their activation products (TNF-a, IL-6, IL-1a, IL-1b, IL-1Ra) into the visceral organs. Moreover, TNF-a together with INF-g and IL-1b induce nitric oxide synthase (NOS) with nitric oxide (NO) production to trigger immunological inflammation in GD.

## 3. Inflammation and Hemostasis

Extensive cross talk exists between inflammation and coagulation, whereby inflammation leads to the activation of coagulation, and coagulation considerably affects inflammatory activity (Figure 1). Proinflammatory cytokines IL-1, IL-6, and TNF-a stimulate the production of tissue factor (TF), a transmembrane glycoprotein that serves as a surface receptor for coagulation factor (F) VIIa. The TF-FVIIa bond plays a key role for the onset of coagulation and thrombin generation [40]. Conversely, activated coagulation proteases may affect specific receptors on inflammatory cells and endothelial cells modulating the inflammatory response [41].

IL-1, IL-6, and TNF-a may also trigger endothelial cells to switch into a procoagulant, clot-promoting state, changing their antithrombotic properties. The same cytokines, together with IL-8 and IL-12, cause platelet activation and clumping [42,43]. In addition, upregulated TNF-a may induce complement component 3 (C3). TNF-a and C3 interact with receptors on platelets. TNF-a, via tumor necrosis factor receptors 1 and 2 (TNFR1 and TNFR2), causes platelet consumption and activation through arachidonic acid pathway stimulation. C3 also interacts with the surface of activated platelets, as well as with other components of the complement system. Platelets may also interact with the complement system through non-classical complement receptor proteins, such as P-selectin or GP1ba. [44]. Finally, the complement plays also an additional role in fibrin deposition [45]. In experimental and clinical GD, a direct relationship between the accumulation of glucosylceramide and the C5a and the C5a receptor 1 axis has been observed [46].

In turn, activated platelets express and release several pro- and anti-inflammatory molecules, which recruit and capture circulating leukocytes and direct them to inflamed tissues. Platelets can also influence adaptive immune responses via the secretion of CD40 and CD40L molecules. Platelet membrane CD40/CD40L allows interaction with different immune cells. Platelet CD40L stimulates the release of IL-8 and MCP-1 from endothelial cells and recruits leukocytes to generate platelet–leukocyte aggregates in areas of vascular inflammation via the expression on platelet surface of adhesion molecules (E-selectin, VCAM, and ICAM-1). Moreover, platelet CD40L induces isotype switching in B-cells and enhances CD8+T-cell responses. Platelets also express CD40 and integrin aIIbb3, receptors for CD40L, by creating feedback loops from the bond.

In addition, platelet granules contain several chemokines and cytokines. CXCL4 or platelet factor 4 (PF4) together with CCL5 enhance the arrest of MOs on endothelial cells and promote their survival, chemotaxis, and differentiation. CXCL7 plays a key role in the recruitment of neutrophils. Platelet release of PF4 and sCD40L upregulates the release of co-stimulatory molecules and proinflammatory cytokines from the dendritic cells. Platelets can directly interact with dendritic cells via CD62P/CD61P, influencing their maturation and enhancing the synthesis of the Th2 helper chemokine CCL17 [47].

## 4. Glycolipids and Hemostasis

Glycolipids are important components of cell membranes and can play critical roles as bioregulators of different processes, such as blood coagulation. Different membrane phospholipid components differently affect coagulation and anticoagulation reactions. For instance, anionic phospholipidis, mainly phosphatidylserine, increase prothrombinase activity; in contrast, posphatidylethanolamine and cardiolipin enhance the activated protein C (APC) anticoagulant pathway.

Thrombin generation is inhibited by glucosylsphingosine, but not by glucosylceramide [48]. In turn, glucosylceramide enhances inactivation of factor Va by APC and protein S, which could represent a potential risk factor for venous thromboembolism [49,50].

It clearly appears that the links between the accumulation of glucosylceramide and glucosylsphingosine, inflammation, and hemostasis in GD may be many and different. At present, however, these are more theoretical links, of which the clinical impact on hemostasis has yet to be outlined. Figure 1 summarizes the relationship between Gaucher cell accumulation, proinflammatory cytokines, and hemostasis.

Gaucher cell accumulation represents the pathogenetic first step of Gaucher disease. On the one hand, it causes direct tissue damage by infiltration; on the other hand, it is responsible for inflammatory cascade activation, with cytokines and chemokines secretion with biological activity and influence on clinical manifestation. IL-1b, TNF-a, IL-6, and IL-10 may contribute to osteopenia; IL-6 and IL-10 may be responsible for gammopathies and multiple myeloma; IL-1b, TNF-a, and IL-6 may have a role in the activation of hypermetabolism. There is also an extensive cross talk between inflammation and hemostasis. On primary hemostasis, proinflammatory cytokines IL-1, IL-6, and TNF-a may trigger endothelial cells to change their antithrombotic properties into a procoagulant, clot-promoting state. The same cytokines, together with IL-8 and IL-12, cause platelet activation and clumping. TNF-a may also induce complement component 3 (C3) and both induce platelet consumption and activation through arachidonic acid pathway stimulation. IL-1, IL-6, and TNF-a stimulate the production of tissue factor (TF), which serves as a surface receptor for coagulation factor (F) VIIa triggering the extrinsic coagulation pathway. The TF–FVIIa bond plays a key role for the onset of coagulation and thrombin generation. The complement system, induced by TNF-a, plays a role in fibrin deposition. The accumulation of glucosylceramide and glucosylsphingosine also has a direct action on hemostasis. In fact, thrombin generation is inhibited by glucosylsphingosine and glucosylceramide can enhance the inactivation of factor Va by APC and protein S.

## 5. GD and Bleeding

Clinically, it is known that a bleeding tendency is one of the early symptoms of GD, being described in around 20% of patients [51]. Bleeding is rarely severe and usually mucocutaneous (epistaxis, gingival bleeding, menorrhagia, bruises). However, post-operative [52,53,54], delivery, and post-partum bleeding [55,56] have been also reported. In addition to these typical bleeding manifestations of primary hemostasis defects, hemorrhages characteristics of coagulation defects, such as spontaneous Iliopsoas hematomas, have also been reported [57].

Several and often concomitant causes can be responsible for hemorrhagic diathesis.

### 5.1. Thrombocytopenia

At diagnosis, reduced platelet count is present in 60% of type 1 GD adult patients, with severe thrombocytopenia (platelets < 60 × 10^9^/L) in 15%, moderate thrombocytopenia (platelets > 60 and <120 × 10^9^/L) in 45%, and mild thrombocytopenia (platelets > 120 and <150 × 10^9^/L) in 40% [6]. Regarding the pediatric population, thrombocytopenia is described at diagnosis in half of the cases [58]. Different and frequently associated causes are responsible for thrombocytopenia. Primitive thrombocytopenia usually occurs as a consequence of impairment of megakaryopoiesis associated with bone marrow infiltration by Gaucher cells. The most frequent etiology of secondary thrombocytopenia is hypersplenism, because splenomegaly is present at presentation in approximately 90% of patients and can be massive [6]. The spleen volume can reach 1500–3000 cc in size, compared to an average 50–200 cc in the normal adult. The shrinkage of spleen volumes and increase of platelet count usually occur within 6 months from the beginning of ERT [17,59] or within 9 months from beginning SRT [27]. Low platelet count in patients with massive splenomegaly may require longer periods before increasing, but continuous improvement usually occurs within the first 2–4 years of treatment. Persistent thrombocytopenia in GD patients treated for over 4 years is associated with refractory splenomegaly. More severe thrombocytopenia and a poor response to ERT may also be associated with the presence of focal splenic lesions [60].

Persistently low platelet counts or a rapid platelet decline [61] may sometimes also be due to the coexistence of immune thrombocytopenia (ITP), not surprising considering the increased incidence of immunological disorders in GD patients. In patients with GD and ITP the use of corticosteroids, high-dose intravenous immunoglobulin, rituximab, and thrombopoietin receptor analogues should be considered as therapeutic options in association with ERT or SRT [62,63].

### 5.2. Platelet Function Defects

Bleeding can occur in GD patients with platelet counts >100 × 10^9^/L and normal coagulation assays, thus suggesting the presence of platelet function abnormalities. Deficient platelet adhesion associated with a history of mucosal bleeding has been reported in two thirds of 48 type 1 GD patients [64] (Table 2). Patients receiving ERT had platelet counts or platelet adhesion no higher than those untreated, suggesting an intrinsic platelet abnormality in GD. The mechanism of reduced platelet adhesion in patients with type 1 GD is unclear, but the increased plasma levels of glucocerebroside may affect platelet activation [65,66,67], as described in two patients with an acquired pseudo-Bernard–Soulier syndrome [68]. Bernard–Soulier syndrome (BSS) [69] is a rare autosomal recessive platelet function disorder caused by the reduction or abnormality of platelet GPIb-IX-V complex which mediates the binding of von Willebrand factor to platelets, thus promoting platelet adhesion to the subendothelium. In patients with BSS, the bleeding time is markedly prolonged, platelet count is moderately decreased, and platelet size largely increased on the peripheral smear. In platelet aggregation tests, the response to the commonly used agonists ADP, epinephrine, collagen, and arachidonate are normal, whereas the response to ristocetin is decreased or absent, similar to what is observed in patients with von Willebrand disease.

An acquired pseudo-Bernard–Soulier syndrome [68] has been diagnosed in two patients with GD and a history of severe bleeding. Both patients were splenectomized, had normal platelet count, and a markedly prolonged bleeding time (>15 min). Laboratory tests showed absent platelet aggregation to ristocetin, associated with normal expression of GPIb-IX-V and VWF levels, suggesting that glucocerebroside accumulating in plasma and coating the platelet membrane could be responsible for the inhibition of ristocetin-induced platelet agglutination. Defective platelet aggregation in response to the main agonists has been reported in approximately 22–61% of case series of patients with type 1 GD (Table 2); like platelet count, aggregation usually improves after starting ERT [52,70,71,72,73].

The analysis of platelet function can be difficult in thrombocytopenic patients. For this reason, in a recent study, flow cytometry has been used [74] to provide a quantitative assessment of platelet function, correlating the finding with bleeding and GD-related data. In 149 patients, a reduced platelet reactivity was found in 53, and 6.7% of them had a more severe platelet dysfunction. A platelet degranulation defect but no αIIbβ3 integrin activation defect seems to be associated with clinical bleeding. More severe platelet dysfunction is associated with a higher level of lyso-Gb1, suggesting again an acquired thrombocytopathy.

### 5.3. Coagulopathy

Multiple clotting abnormalities of variable severity have been reported in GD patients. The detection of prolongations of PT and/or aPTT must prompt assaying the plasma levels of single coagulation factors. In one of the largest studies including 30 type 1 GD patients, at diagnosis, PT was prolonged in 42% and APTT in 38% of them [75]. The most common coagulation factor deficiencies are factor V, factor X, and factor II, but the reduction of all circulating clotting factors has also been reported [53,71,72,73,75,76,77,78,79,80] (Table 3). Variable deficiency of clotting factors is observed at diagnosis, with levels as low as 15% in some patients [75]. ERT has been shown to improve or normalize the clotting factor levels in plasma [71,81]. A high frequency of isolated factor XI deficiency has been reported in some patients with GD [72,82,83], suggesting an association between the two diseases. However, a poor correlation between FXI level and the severity of bleeding has been shown. Thus, since factor XI deficiency has a high prevalence in the Ashkenazi Jewish population, with a heterozygosity as high as 9%, when present, this association is likely casual.

The pathogenetic mechanisms responsible for clotting factor deficiencies in GD are various and different. Liver disease may reduce the synthesis of clotting factors, an enlarged spleen may result in their increased clearance, and the elevated concentration of circulating glucocerebroside or the presence of antiphospholipid antibodies may interfere with the clotting cascade [84]. Moreover, a significant enhancement of coagulation, as suggested by elevated concentration of the markers of activation of coagulation (thrombin-antithrombin complex) and fibrinolysis (plasmin-α2antiplasmin complexes, fibrin cleavage product D-dimer), has been described, especially in splenectomized patients. Thus, clotting factor defects may also be linked to their consumption caused by an ongoing low-grade intravascular coagulation and fibrinolysis, possibly triggered by IL-1α, TNF-α, and IL-6 secreted by the Gaucher cells [75].

## 6. Hemostatic Management

Even though ERT and SRT may improve and correct hemostatic abnormalities in GD, treatments meant to enhance hemostasis may be necessary in specific clinical circumstances, such as recent onset of ERT/SRT, surgery, delivery, and life-threatening bleeding.

A laboratory screening assessment is required, with an evaluation of platelet count and platelet function tests, i.e., platelet function analyzer (PFA-100) [85] and/or platelet aggregation, APTT, and PT, in order to provide the most appropriate hemostatic treatment. When an abnormality of coagulation screening test is found, a diagnostic deepening may be recommended. After the exclusion of FVIII, FIX, and FXII defects, an isolated prolonged APTT suggests FXI deficiency, and an isolated PT prolongation is typically caused by a FVII deficiency. The combined prolongation of APTT and PT suggests the diagnosis of combined FV + FVIII, FX, FV, FII, or fibrinogen deficiencies.

Thromboelastography (TEG) may be an alternative option to evaluate the bleeding tendency in a peri-surgical or predelivery setting [86]. TEG provides a quick and visualized monitoring and analysis of the viscoelastic properties of clot formation and dissolution in whole blood. In addition, the separate effects of platelets and fibrinogen on overall clot strength can be rapidly assessed [87]. The goal of TEG is the evaluation at the point of care for the management decision of the need for blood transfusion or fibrinogen supplementation. However, the experience of using TEG in patients with GD is still limited and the assay is not performed in a sufficiently uniform manner worldwide [88].

## 7. Hemostatic Agents

Several therapeutic options to enhance an impaired hemostatic function are available, ranging from antifibrinolytics to replacement therapy with platelet transfusion or clotting factor concentrates. **Tranexamic acid** represents an important therapeutic aid for the management of mucocutaneous bleeding [89]. This synthetic derivative of the amino acid lysine exerts its antifibrinolytic effect through the reversible blockade of lysine binding sites on plasminogen molecules. It is a cheap and easy to use drug for the treatment and prevention of bleeding. Tranexamic acid is used at a dose of 15–25 mg/kg every 8 h for 3–6 days by oral or intravenous administration for bleeding or surgery. In addition, it can be used as mouthwash for oral bleeding or dental procedures. No evidence of increased thromboembolic or other significant adverse events has been reported in different large populations treated with tranexamic acid.

To control gynecological bleeding, **hormone therapy** alone or in combination with tranexamic acid is often used [56,90]. **Oral contraceptives** containing both progestin and estrogen should be started first. In women who failed to have a clinically useful response, a levonorgestrel-releasing intrauterine device may be recommended, such as in women with inherited bleeding disorders [91].

When tranexamic acid alone is not able to guarantee an effective hemostasis, **desmopressin** may be employed. Desmopressin (DDAVP, 1-deamino-8-D-arginin-vasopressin) is a synthetic analog of the antidiuretic hormone vasopressin, V2 agonist, able to increase VWF and FVIII in plasma, and, therefore, it is the drug of choice for mild hemophilia A and type 1 von Willebrand disease [92]. DDAVP is also clinically efficacious in patients with other, generally mild platelet function disorders of the [93,94,95,96,97], and its use is recommended for bleeding prophylaxis for situations at low risk of bleeding risk [98]. DDAVP-induced the rise of circulating VWF high-molecular-weight multimers, leading to an increased platelet adhesion to the injured vessel wall, which could enhance hemostasis in patients with platelet function disorders and in those with GD [99,100]. In addition, in vivo DDAVP strengthens the ability to form procoagulant platelets and increases platelet-dependent thrombin generation by inducing intracellular Na^+^ and Ca^2+^ fluxes [101]. DDAVP can be administered intravenously (0.3 µg/kg diluted in 50–100 mL saline and infused over 15–30 min), subcutaneously at the same dose, or intranasally (fixed doses of 300 µg in adults and 150 µg in children). Subsequent doses may be administered every 12–24 h, but a progressive reduction in magnitude of the response (tachyphylaxis) is observed after closely-spaced doses. Side effects, due to the vasomotor effect of the molecule, may include mild tachycardia, flushing, and headache. Rare side effects attributable to the antidiuretic properties of DDAVP are hyponatremia, especially in children below the age of 2, and volume overload. DDAVP should be used with caution in elderly subjects with atherosclerotic disease [102].

In GD patients with severe thrombocytopenia and/or severe functional defects, **platelet transfusion** may be recommended, as well as specific **clotting factor concentrates** administration when a serious coagulation defect is diagnosed [103]. Figure 2 summarizes the therapeutic options to enhance hemostasis and their mechanisms of action.

Several agents with action on different hemostatic phases are available. In primary hemostasis, platelet transfusion increases platelet number and improves their function, whereas desmopressin increases von Willebrand factor and FVIII and improves platelet adhesion. In the coagulation pathway, desmopressin increases FVIII, whereas clotting factor concentrates increase levels of corresponding coagulation factors (e.g., FII, FVII, FIX, FX, FVIII).

In the fibrinolysis, tranexamic acid blocks plasmin generation and can be administered alone or with all the other treatments.

## 8. Discussion

Type 1 GD is a rare disease with heterogeneous multisystem involvement, showing different severity of symptoms at each age. In recent years, it has been understood that the pathophysiology is caused not just by the burden of glucosylceramide and glucosylsphingosyne storage, but by macrophage activation and the resulting cascade of inflammatory events. The contribution of definite proinflammatory cytokines is emerging more and more clearly in the pathogenesis of precise clinical manifestations in GD. IL-1b, TNF-a, IL-6, and IL-10 may contribute to osteopenia; IL-1b, TNF-a, and IL-6 may have a role in the activation of coagulation and hypermetabolism; IL-6 and IL-10 may be responsible for gammopathies and multiple myeloma. Similarly, more and more cross-links between inflammation and hemostasis are emerging, such as some direct roles of glucosylceramide and glucosylsphingosyne on coagulation and anticoagulation reactions. These interactions may suggest that the prothrombotic risk in GD is high; the studies on thrombophilic profiles have not been conclusive [104]. The indicator of active microvascular thrombosis, D-dimer, was found to be related with severity of bone and lung involvement [105,106], though this could imply that microthrombi are part of the etiology for avascular necrosis and pulmonary hypertension in patients with GD.

A bleeding tendency has been reported frequently, particularly at diagnosis. The abnormalities of the hemostatic pathways in GD concern both primary hemostasis and coagulation. Thrombocytopenia, platelet dysfunction, and clotting factor deficiencies may be variously associated with each other, predicting a hemorrhagic risk. Thrombocytopenia can have several and often concomitant etiologies, such as bone marrow infiltration by Gaucher cells with impairment of megakaryopoiesis, hypersplenism, and autoimmune. Platelet function defects are likely to be related to glucosylceramide and glucosylsphingosine accumulation in plasma with consequent platelet membrane coating and the blocking of glycoproteins for adhesion and aggregation. In support of this theory, there is the improvement of platelet function with ERT. Glucosylceramide and glucosylsphingosine excess also interfere with coagulation through thrombin generation inhibition and anticoagulant APC/PS system activation. Clotting factor deficiencies may also be due to reduced synthesis for hepatic disease, increased clearance in enlarged spleen, or accelerated consumption in a state of low-grade intravascular coagulation and fibrinolysis, supported by IL-1α, TNF-α, and IL-6.

ERT and SRT are effective in improving hematological laboratory parameters and symptoms, including bleeding. However, there are situations in which the hemorrhagic risk can be significant in GD patients. Special attention should be paid in the early stages of ERT/SRT, at surgery delivery, and when life-threatening bleeding occur. A careful assessment of the bleeding risk is always advisable before invasive procedures or delivery, since several treatment options are available for the prevention of bleeding in GD patients, ranging from tranexamic acid to DDAVP and platelet transfusion or clotting factor concentrates.

In conclusion, the present review was aimed at raising attention to a rarely addressed clinical problem associated with GD, highlighting the complex pathophysiological interplay between hemostasis and inflammation caused by a lysosomal storage disease and the therapeutic options available to manage bleeding complications.

## Figures and Tables

**Figure 1 jcm-11-06920-f001:**
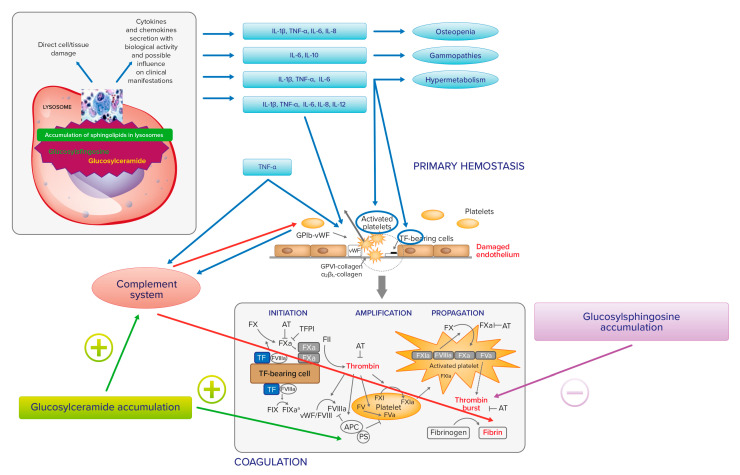
Relationship between Gaucher cell accumulation, proinflammatory cytokines, and hemostasis.

**Figure 2 jcm-11-06920-f002:**
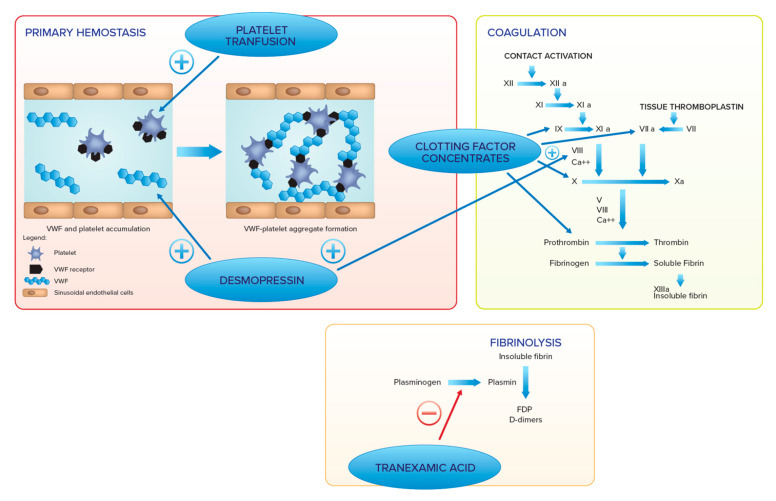
Therapeutic options to enhance hemostasis.

**Table 1 jcm-11-06920-t001:** Clinical subtypes of Gaucher disease.

Subtype	Incidence	Ethnic Group	Age at Onset	Primary CNS Involvement	Splenomegaly and/or Hepatomegaly	Cytopenia ^1^	Bone Disease ^2^	Other
Type 1	1:40,000–1:60,000 1:450 in Ashkenazi Jews	Panethnic, more common in Ashkenazi Jews	Any age	No	Yes	Yes	Yes	Pulmonary disease
Type 2 (Acute or infantile)	<1:100,000	Panethnic	Infancy—early childhood	Bulbar signs Pyramidal signs Cognitive impairment	Yes	Yes	No	Pulmonary disease Dermatologic changes
Type 3 (Subacute, juvenile)	<1:50,000 to <1:100,000	Panethnic	Childhood	Oculomotor apraxia Seizures Progressive myoclonic epilepsy	Yes	Yes	Yes	Pulmonary disease
Perinatal lethal form	<1:10,000	Panethnic	Perinatal	Pyramidal signs	No	No	No	Ichthyosiform or collodion skin changes Nonimmune hydrops fetalis

^1^ Thrombocytopenia, anemia, leukopenia; ^2^ osteopenia, focal lytic or sclerotic lesions, osteonecrosis.

**Table 2 jcm-11-06920-t002:** Studies of platelet abnormalities in Gaucher disease before start ERT/SRT: percentage of patients in each study found to have bleeding and platelet function abnormalities.

Reference	Gillis et al., 1999 [70]	Giona et al., 2006 [71]	Spectre et al., 2011 [64]	Mitrovic et al., 2012 [72]	Komninaka et al., 2020 [73]	Revel-Vilk et al., 2021 [74]
Patients (N)	32	13	48	31	29	149
Bleeding manifestations	12.5%	15%	58%	32%	79%	49%
Plts (×10^9^ L^−1^) (median; range)	180 (74–508)	142 (51–284)	193 (153–252)	108	147 (103–192)	176 (30–485)
Plt adhesion deficiency	-	-	66%	-	-	-
Plt aggregation abnormalities	22%	46%	22%	61%	-	-
PFA Collagen/EPI prolonged PFA Collagen/ADP prolonged PFA Collagen/EPI + Collagen/ADP prolonged	- - -	- - -	- - -	- - -	55% 79% 52%	- - -
Platelet reactivity deficiency	-	-	-	-	-	53%

Legend: Plts, platelets; PFA, platelet function analyzer.

**Table 3 jcm-11-06920-t003:** Studies of coagulation factor abnormalities in Gaucher disease: percentage of patients in each study found to have coagulation abnormalities.

Reference	Boklan et al. (1976) [76]	Billett et al. (1996) [77]	Hollak et al. (1997) [75]	Katz et al. (1999) [53]	Barone et al. (2000) [78]	Giona et al. (2006) [71]	Deghady et al. (2006) [79]	Mitrovic et al. (2012) [72]	Serratrice et al. (2019) [80]	Komninaka et al. (2020) [73]
Patients (N)	11	9	30	28	5	15	10	31	43	29
↑ PT	-	11%	42%	81%	80%	47%	100%	52%	12%	0%
↑ APTT	100%	55%	38%	-	100%	53%	60%	42%	21%	7%
↑ Fibrinogen	-	0	-	-	-	-	-	-	0	7%
↓ FII	-	11%	50%	-	20%	-	20%	-	8%	-
↓ FV	18%	22%	87%	27%	100%	23%	30%	-	31%	-
↓ FVII	9%	-	33%	-	0	8%	50%	-	3.4%	-
↓ FVIII	27%	11%	10%	27%	60%	15%	30%	-	4%	0
↓ FIX	73%	0	3%	14%	20%	31%	20%	-	4.50	-
↓ FX	9%	-	57%	-	20%	8%	10%	-	11.50	-
↓ FXI	-	33%	27%	36%	40%	0	0	-	20%	-
↓ FXII	-	-	30%	27%	25%	15%	10%	-	13%	-
↓ VWF	-	11%	-	-	-	0	-	-		0
↓ ADAMST13	-	-	-	-	-	-	-	-	0	48%
↓ PC	-	-	26%	-	-	-	-	10%	11.5%	0
↓ PS	-	-	11%	-	-	-	-	-	19%	0
↓ AT	-	-	3%	-	-	-	-	0	0	0
↑ D dimer	-	-	68%	-	-	-	-	76%	-	38%
↑ TAT	-	-	46%	-	-	-	-	43%	-	-
↓ PAI	-	-	-	-	-	-	-	-	-	69%

↑ = prolonged/increased; **↓ =** reduced.

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
