# Peer review of "Hemostatic Abnormalities in Gaucher Disease: Mechanisms and Clinical Implications"

_jcm, 2022, doi:10.3390/jcm11236920_

Round 1

Reviewer 1 Report (Previous Reviewer 1)

Congratulations for the nice paper. 

Author Response

We thank the reviewer for appreciating our paper!

Reviewer 2 Report (New Reviewer)

The presented review is interesting and contains the vast of essential data in the field of Gaucher diseases.

However, I found a few issues to correct in the manuscript.

1.       When describing inflammation and hemostasis, link it to your Fig. 1. There is no linking to the figure in the text.

2.       Adding abbreviations list at the beginning would be useful.

3.       Please check for editing, like in line: 288 there is pseudo-pseudo…. Is it an error? In the table 2 some parts of the text is all merged like plateletsadhesiondeficinecy.. etc

4.       Cite table 2 in the text.

5.       Line 324-325: Please clarify: If there is ‘’…. poor correlation between FXI level and severity of

suggesting an association between the two diseases’’ Is it really? If there is poor correlation it sound like there is no or very little association.

6.       Table 3: Cite table 3 and figure 2 in the manuscript. For Table 3 there is the same description at the top and the bottom of the table.

7.       In the discussion part: In the final sentences just make it clear why the review is novel and important in the field.

Author Response

We thank the reviewer for useful comments. As to her/his comments, please note the following considerations:

-When describing inflammation and hemostasis, link it to your Fig. 1. There is no linking to the figure in the text.

R: This has now been done

-Adding abbreviations list at the beginning would be useful.

R: Done

- Please check for editing, like in line: 288 there is pseudo-pseudo…. Is it an error? In the table 2 some parts of the text is all merged like plateletsadhesiondeficinecy.. etc

R: Done

-Cite table 2 in the text.

R: Done

-Line 324-325: Please clarify: If there is ‘’…. poor correlation between FXI level and severity of suggesting an association between the two diseases’’ Is it really? If there is poor correlation it sound like there is no or very little association.

R: The sentence has now been rephrased

-Table 3: Cite table 3 and figure 2 in the manuscript. For Table 3 there is the same description at the top and the bottom of the table.

R: Done

-In the discussion part: In the final sentences just make it clear why the review is novel and important in the field.

A sentence has now been added at the end of discussion section

Please find in red the more significant changes to the manuscript. We have also checked for typos and overlapping with previous publications.

We hope that now the paper is usitable for publication in JCM.

With kind regards,

Giancarlo Castaman

This manuscript is a resubmission of an earlier submission. The following is a list of the peer review reports and author responses from that submission.

Round 1

Reviewer 1 Report

Congratulations on this nice paper.

My suggestions and questions are as follow:

1. I suggest using the following description of Gaucher disease: GD is an autosomal recessive lysosomal storage disease that is caused by deficiency of the enzyme glucocerebrosidase which is required for the degradation of glycosphingolipids.

2. In addition to thrombocytopenia, there are other haematological manifestations of GD; anaemia (due to hypersplenism, hepcidin dysregulation, and impaired erythropoiesis), gammopathies (mono- or polyclonal, multiple myeloma). Did you assess these abnormalities?

3. What's the authors' explanation of defective platelet activation, adhesion, membrane function, and increased fibrinolysis?

4. Any data about the changes in the haemostasis function of GD patients after ERT?

5. Was there a measurement of serum erythropoietin level?

6. Any data about the serum ferritin level in this group of patients?

7. Was there any correlation between factor XI level and the degree of bleeding?

8. The discussion section is very short and does not reflect the authors' nice work. highly recommend 

Reviewer 2 Report

The article exemplifies the mechanisms involved in the occurrence of hemorrhages in patients with DG in different clinical situations: thrombocythemia, coagulopathies, platelet function defects. There is also a description of the forms of action of the main hemostatic agents: tranexamic acid, hormone therapy, oral contraceptives, desmopressin, platelet transfusion, specific clotting factor concentrates administration. However, I suggest that the authors draw up schematic figures to explain the different mechanisms involved in the occurrence of hemorrhages, as well as the mode of action of the therapies used to reduce hemorrhage in DG in different clinical situations.